# Conserved Evolution of MHC Supertypes among Japanese Frogs Suggests Selection for Bd Resistance

**DOI:** 10.3390/ani13132121

**Published:** 2023-06-27

**Authors:** Quintin Lau, Takeshi Igawa, Tiffany A. Kosch, Anik B. Dharmayanthi, Lee Berger, Lee F. Skerratt, Yoko Satta

**Affiliations:** 1Research Center for Integrative Evolutionary Science, Sokendai (The Graduate University for Advanced Studies), Hayama 240-0115, Japan; quintin@soken.ac.jp; 2Amphibian Research Center, Hiroshima University, Higashi-Hiroshima 739-8526, Japan; 3One Health Research Group, Faculty of Science, University of Melbourne, Parkville 3010, Australialee.berger@unimelb.edu.au (L.B.);; 4Research Center for Biosystematics and Evolution, National Research and Innovation Agency (BRIN), Bogor 16911, Indonesia

**Keywords:** anuran, major histocompatibility complex, chytrid fungus, *Batrachochytrium dendrobatidis*, MHC supertyping, NetMHCIIpan, MHC peptide binding

## Abstract

**Simple Summary:**

This study explored the major histocompatibility complex (MHC) of a variety of Japanese frog species, to further understand why they may have resistance to the deadly chytrid fungus *Batrachochytrium dendrobatidis* (Bd), which causes mortality and decline in many other amphibians. MHC supertyping analysis showed that all examined East Asian frogs contained at least one MHC-IIb allele belonging to supertype ST-1, indicating that functional properties in the peptide binding sites of MHC-II are conserved among East Asian frogs and some other anurans across the world. Preliminary analysis also suggests that MHC-IIb supertypes ST-1 and ST-2 have higher overall peptide binding ability, regardless of where the peptides are derived from (both Bd and non-Bd). The findings support the hypothesis that MHC-IIb among East Asian frogs may have co-evolved under similar selective pressures, possibly due to the presence of Bd in the region, contributing to their resistance to the disease. Our study helps to further elucidate the complex relationships between anuran MHC and Bd pathogen.

**Abstract:**

The chytrid fungus *Batrachochytrium dendrobatidis* (Bd) is a major threat to amphibians, yet there are no reports of major disease impacts in East Asian frogs. Genetic variation of the major histocompatibility complex (MHC) has been associated with resistance to Bd in frogs from East Asia and worldwide. Using transcriptomic data collated from 11 Japanese frog species (one individual per species), we isolated MHC class I and IIb sequences and validated using molecular cloning. We then compared MHC from Japanese frogs and other species worldwide, with varying Bd susceptibility. Supertyping analysis, which groups MHC alleles based on physicochemical properties of peptide binding sites, identified that all examined East Asian frogs contained at least one MHC-IIb allele belonging to supertype ST-1. This indicates that, despite the large divergence times between some Japanese frogs (up to 145 million years), particular functional properties in the peptide binding sites of MHC-II are conserved among East Asian frogs. Furthermore, preliminary analysis using NetMHCIIpan-4.0, which predicts potential Bd-peptide binding ability, suggests that MHC-IIb ST-1 and ST-2 have higher overall peptide binding ability than other supertypes, irrespective of whether the peptides are derived from Bd, other fungi, or bacteria. Our findings suggest that MHC-IIb among East Asian frogs may have co-evolved under the same selective pressure. Given that Bd originated in this region, it may be a major driver of MHC evolution in East Asian frogs.

## 1. Introduction

The major histocompatibility complex (MHC) is a vital component of the vertebrate adaptive immune system. MHC genes encode important glycoprotein receptors that recognize, bind, and present specific antigens derived from pathogens to T lymphocytes. There are two major classes of MHC: class I (MHC-I) and class II (MHC-II). MHC-I molecules are ubiquitously expressed on most nucleated cells and predominantly present endogenous antigenic peptides from intracellular pathogens (e.g., viruses); the α1 and α2 domains (heavy chain) form the peptide binding platform and surface for recognition by cytotoxic CD8+ cells [1,2]. MHC-II molecules are found on professional antigen-presenting cells (including B lymphocytes, dendritic cells, macrophages, and in most species, activated T lymphocytes) and present exogenous antigenic peptides from extracellular pathogens (e.g., many bacteria and fungi) to CD4+ T helper cells. They are formed from a heterodimer of alpha and beta chains, and the open-ended peptide-binding region or groove (PBR) comprises the α1 and β1 domains [2].

Many associations have been identified between vertebrate MHC genes and resistance to specific pathogens, whether it be specific MHC alleles or heterozygous states of MHC allelic combinations [3,4,5]. In addition, structural features of the MHC PBR, which is determined by variable amino acid sites, may alter the affinity for specific peptides and determine the population of peptides that are recognized, bound, and presented [6,7]. Important binding pockets include anchor residues P2 or P5/6 and PΩ in MHC-I, and P1, P4, P6, and P9 in MHC-II [7]. MHC class I peptide binding regions can usually accommodate peptides of around 8–10 residues [8], whereas MHC class II can accommodate 13–25 residues in their open binding groove [9].

In wild frogs, commonly identified infectious diseases include ranaviruses and chytridiomycosis, caused by *Batrachochytrium dendrobatidis* (Bd fungus) [10]. The Bd fungus disrupts amphibian skin, and has caused mortality and population declines, as well as the extinction of about 90 species in the last 30 years [11]. Based on whole-genome sequencing of the global diversity of Bd isolates, East Asia was identified as the originating source of Bd [12]. A more recent study discovered a divergent line of Bd that is endemic to Asia [13], thereby further supporting that Bd originated in Asia. Experimental infections using various Bd strains showed that *Bufo gargarizans*, *Bombina orientalis*, and *Hyla* (or *Dryophytes) japonica* from South Korea can clear the fungus, whereas a susceptible Australian species *Litoria caerulea* succumbs to Bd [14], supporting the hypothesis that frogs from East Asia are resistant. Moreover, the prevalence of Bd in Japan is low [15] and there are no published reports of Bd-related susceptibility or death in Japanese frogs. Bd endemism in East Asia likely provided ample time for host-pathogen evolution, and this may explain the present lack of Bd mortality and morbidity among frogs from this region (including Japanese frogs).

While resistance to infectious pathogens, such as Bd, is most likely polygenetic, the MHC is thought to play a key role, with an increasing number of studies supporting MHC being correlated with Bd resistance [16,17,18,19,20]. Bataille et al. [16] found that MHC-II conformation was associated with Bd resistance. They showed that amino acid properties at P9 binding pockets were conserved between resistant Korean frog species (*B. gargarizans* and *B. orientalis*) and surviving individuals of a susceptible Australian species (*Litoria verreauxii alpina*). In a Bd-challenge study in the highly endangered *Pseudophryne corroboree*, another Australian species susceptible to Bd, three class I MHC variants and one MHC supertype were found to be associated with survival and Bd infection load [17]. Studies in wild and captive North American frogs have also identified associations between specific MHC-IIb (MHC-II β1 domain) alleles or supertypes with Bd susceptibility or resistance [18,19]. A more recent study in *Rana pipiens* in North America found MHC-II supertypes associated with (significant) elevated Bd infection risk or (nearly significant) reduced Bd risk, and that MHC heterozygosity significantly explained spatial patterns of Bd prevalence [20]. These studies have highlighted that MHC-based genetic information are important for informing breeding programs aimed at assisting species recovery and conservation in the face of Bd-driven declines.

MHC supertyping is an approach whereby alleles are grouped, using principal component analyses, based on physicochemical properties of amino acid sites in the PBR. Recently, we also used supertyping analyses to investigate MHC-IIb of three Japanese brown frog species (*Rana japonica, Rana ornativentris,* and *Rana tagoi tagoi*) along with other publicly available frog species, and found initial evidence supporting an MHC-II supertype shared among the three Ranidae species from Japan as well as two species from Korea (*B. gargarizans* and *B. orientalis*), all of which appear resistant to Bd. Nevertheless, the MHC of many other East Asian amphibians remains unexplored. Notably, the Japanese archipelago boasts a remarkable anuran species diversity of 52 native species, despite its limited land area. Thus, to comprehensively understand the association between MHC diversity and chytrid fungi resistance, a diverse range of potentially resistant species needs to be compared.

Therefore, the aim of the present study was to expand the study of MHC genes across a diverse range of common (widely distributed) Japanese frog species and compare them to MHC from Bd-resistant and susceptible species, which will further contribute to understanding amphibian-chytridiomycosis dynamics. We hypothesize that the MHC plays an important role in the resistance of Japanese frogs to pathogens such as Bd. Using phylogenetic, supertyping, and preliminary MHC-function prediction analyses, we examine if there are any evolutionary and functional features of MHC genes conserved among Japanese frog species that may contribute to immune resistance against pathogens including Bd.

## 2. Materials and Methods

### 2.1. Animals, Nucleic Acid Isolation, Next-Generation Sequencing, and De Novo Assembly

In the present study, transcriptomic datasets were produced for eight Japanese frog species: one Bufonidae species (*Bufo japonicus*); one Dicroglossidae species (*Fejervarya kawamurai*); one Hylidae species (*Dryophytes japonicus*); three Ranidae species (*Glandirana rugosa, Pelophylax nigromaculatus,* and *Pelophylax porosus porosus*); and two Rhacophoridae species (*Buergeria buergeri* and *Buergeria japonica*) (Appendix A). Many of the study species are endemic to Japan, and some are also distributed in other East Asian countries: *D. japonicus* in Korea and China, *F. kawamurai* in Taiwan and China, and *P. nigromaculatus* across East Asia.

For each of the species, a single adult individual (captive-reared at the Amphibian Institute in the University of Hiroshima) was euthanized through topical application of tricaine methanesulfonate (MS222) on the dorsal skin. All individuals were kept under a clean and slow water flow aquarium in accordance with the recommendations in the Guide for the Care and Use of Laboratory Animals of Hiroshima University Animal Research Committee (HUARC); all procedures were also approved by HUARC (Approval number: G17-9). Spleen samples were then immediately collected for immediate RNA extraction using ISOGEN (Nippon Gene, Tokyo, Japan) following manufacturer instructions. Sequencing libraries were created from the eight spleen RNA samples (380–1150 ng RNA/sample) using NEBNext^®^ Poly(A) mRNA Magnetic Isolation Module and NEBNext^®^ Ultra™ RNA Library Prep Kit for Illumina^®^ (New England Bio Labs, Ipswich, MA, USA). The libraries then underwent short read DNA sequencing (paired-end 100-bp) with a cDNA Illumina Hiseq2000 sequencing (15 Gb per sample).

De novo transcriptome assembly for each species was conducted in a similar manner to our previous study [21]. In brief, clean short read data were used for de novo short-read transcriptome assembly with default settings in Trinity version 2.6.5 [22]. Then, the TransDecoder program in the Trinity platform was used to obtain open reading frames and amino acid sequences, and assembled transcripts were annotated against four databases [Swissprot protein database (http://www.expasy.ch/sprot); human amino acid sequence dataset GRCh38.p5 from Ensembl (http://www.ensembl.org/); the KO database (http://www.genome.jp/kegg/ko.html); and the GO database (http://www.geneontology.org/); all accessed on 1 April 2018] using NCBI-BLAST-2.4.0 with E-value < 1 × 10^−5^.

In addition to the assembled transcriptome data from the eight species, we also included transcriptome sequence data from three other Japanese *Rana* species (*R. japonica, R. ornativentris,* and *R. tagoi tagoi)* from which we had previously compiled and characterized the MHC [21,23], leading to a total of 11 Japanese frog species (from seven genera) analyzed in the present study.

### 2.2. Isolation of MHC Sequences

Initially, we collected all assembled transcripts that were annotated as MHC class I or II in the NCBI-BLAST search of the transcriptome data. Subsequently, only full coding MHC sequences were retained. These sequences were then confirmed with additional NCBI-BLAST-2.4.0 search for MHC class I and IIb sequences. For search queries, we used *R. japonica* and *Zhangixalus omeimontis* sequences for MHC class I (GenBank accession numbers KX100487.1 and KC261663.1, respectively) and *R. japonica*, *Lithobates catesbeianus*, and *Rhinella marina* sequences (MF555153.1, KY587181.1, and KP995700.1, respectively) for MHC class IIb.

Since the MHC is one of the most polymorphic and duplicated gene regions in the vertebrate genome [24], multiple assembled MHC transcripts were isolated from most of the study species. However, the polymorphic nature of this gene region also increases the possibility of assembly errors; thus, we used polymerase chain reaction (PCR) and molecular cloning and sequencing to validate the MHC variants. For PCR amplification of MHC-I, we aligned assembled transcripts from all species and designed degenerate primers that successfully amplified across all the study species (Table 1). Degenerate primers were unsuccessful for PCR amplification of MHC-IIb variants; thus, transcripts retrieved from the assembly were aligned within each of the species, and species-specific primers were designed using Primer3 [25] (Table 1). We used the same methods as our previous studies for PCR, molecular cloning, and Sanger sequencing methods [21,23].

### 2.3. MHC Phylogenetic Analyses

Phylogenetic relationships were independently examined for MHC class I (546 bp; exons 2–3, covering both α1 and α2 domains) and MHC class II β1 domain (291 bp; exon 2 only, since published full exon 3 sequences were not available for many anurans). Using BioEdit v7.2, nucleotide sequences of MHC class I and IIb variants characterized from Japanese frogs from our previous studies (KX100486-KX100545, MF555153-MF555185) and present study were aligned (separately for each MHC class) with published MHC alleles/variants from other species (for species and accession numbers, refer to Figure 1, Figure 2 and Figure 3 and Appendix A).

Using MEGA X [26], we constructed custom distance matrices (Nei-Gojobori method [No. of differences]; 1000 bootstrap replicates) and neighbor joining trees based on putative neutral sites (non-synonymous substitutions at the non-PBR sites + all synonymous substitutions) [27,28,29]. PBR sites were based on those characterized in human leukocyte antigen (HLA, the human equivalent of MHC) [6,30,31]. We analyzed 38 codon sites for MHC-I and 16 codon sites for MHC-IIb, and the latter has been used previously as anuran MHC-II PBRs for supertyping analyses [16,21].

### 2.4. Supertyping Analyses

We adapted the methods from previous studies [16,17,18,21,32,33] to conduct supertyping analyses separately for MHC-I and MHC-II. Briefly, codons for the PBR were collated including 12 codon sites for MHC-I based on PBR pocket positions [17,34], and 13 codon sites for MHC-IIb that are considered to affect the P4, P6, and P9 pockets of the MHC-II peptide binding groove [6] (see Figure 1).

Five physicochemical descriptor variables were imputed for each site (z1, hydrophobicity; z2, steric bulk; z3, polarity; and z4/z5, electronic effects). All z-values were then used for discriminant analysis of principle components (DAPC) to cluster the alleles based on predicted physicochemical properties using the R package adegenet [35]. Variants were assigned to clusters using the K-means clustering algorithm and selecting model with low Bayesian information criterion.

We then compared MHC (class I or II) variants from the 11 Japanese frogs, which are assumed to be Bd-resistant, with that of other species with varying susceptibility. Additional Bd-susceptibility categorization for the species shown below is generalized based on extrapolation from previous reports [36,37,38,39,40], and used in the present study to examine general associations between MHC supertypes and Bd susceptibility.

The other species included for MHC-I supertyping were: (a) *B. gargarizans* and *D. japonicus* (resistant) from Korea [33]; (b) *Odorrana margaretae*, *Zhangixalus chenfui*, *Polypedates megacephalus*, and *Z. omeimontis* (assumed resistant) from China [41,42,43]; (c) *Agalychnis callidryas*, *Espadarana prosoblepon*, and *Smilisca phaeota* (tolerant/carrier) from Central America; (d) *L. catesbeianus* (tolerant/carrier), *Lithobates clamitans* (tolerant), *Lithobates yavapaiensis* (susceptible), and *Rana pipiens* (susceptible) from North America [44]; (e) *P. corroboree* (susceptible) from Australia [17]; and (f) *Xenopus laevis* (resistant) from Africa.

The other species included for MHC-II supertyping were: (a) *B. orientalis* and *B. gargarizans* (resistant) from South Korea [16]; (b) *L. catesbeiana* (transcriptome data, DRA accession number SRP051787); (c) *L. yavapaiensis* from North America [18]; (d) *Z. omeimontis* from China [45]; (e) representative sequences from *Epidalea calamita* (low susceptibility), *Nanorana parkeri* (susceptibility unclear), *Rhinella marina* (tolerant), and *Quasipaa spinosa* (susceptibility unclear) available in GenBank; and (f) *L. verreauxii alpina* (susceptible) from Australia but with some resistant populations or individuals [16]. MHC-IIb sequences from Chinese populations of *P. nigromaculatus* [46] were initially included but formed a very distinct ST cluster and thereby excluded.

**Figure 1 animals-13-02121-f001:**
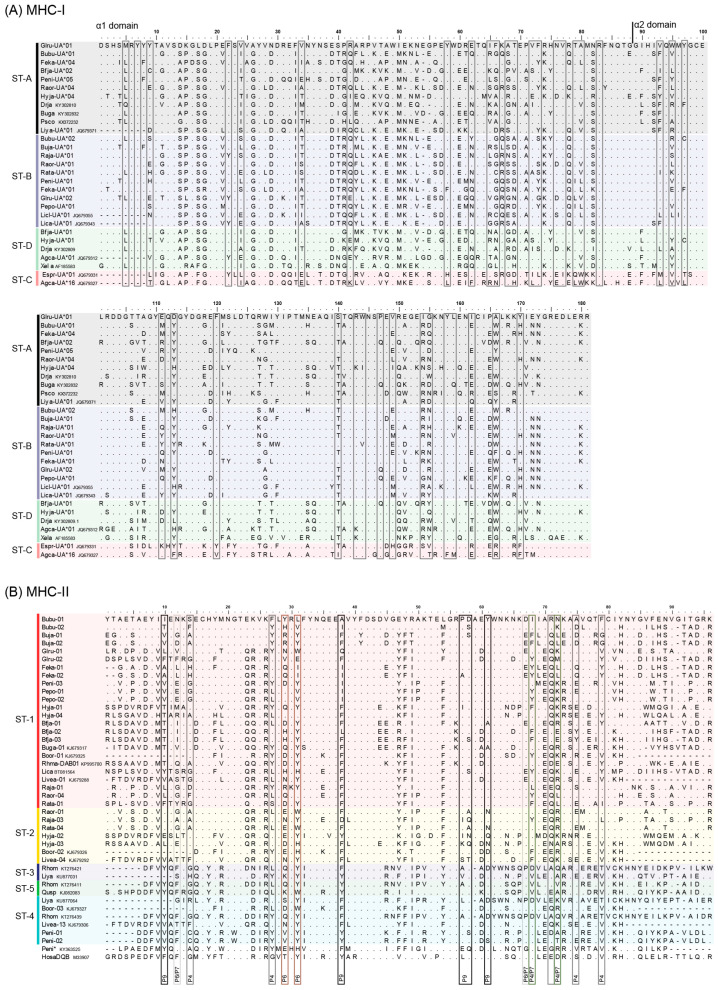
Amino acid alignment of select (**A**) MHC class I [α1 and α2 domain; exons 2–3] and (**B**) MHC class IIb [b1 domain; exon 2] amino acid variants identified in Japanese anuran frog species and other species. Peptide-binding residues are indicated by boxes; for MHC-IIb, pocket residue numbers are indicated at the bottom. Identified MHC supertypes are indicated on the left of the alignments (and colored for MHC-II). See Figure 3 caption for full species names.

### 2.5. Preliminary MHC-II Binding Prediction

We used NetMHCIIpan-4.0, an MHC class II pan-specific method (i.e., can examine any host MHC and peptide sequence) that is trained using quantitative MHC binding data in humans [47] for the preliminary examination of interactive affinity between Bd peptides and anuran MHC. Although application of a similar pan-specific method was demonstrated in non- human primate, pig, and mice MHC-I [48], it is expected that such predictions could also be applied successfully to MHC-II molecules of other organisms. Due to the emphasis on interaction of anuran frog immunity with Bd fungus and our novel finding of a shared MHC-II supertype among all East Asian frogs studied, we focused on MHC class II for peptide binding predictions. For binding prediction using NetMHCIIpan4.0, input data comprised (a) antigenic peptide sequence datasets and (b) anuran MHC-II allele/variant sequences.

*Antigenic peptides:* following our previously used methods [49], we filtered antigenic peptides that were 15–24 amino acids in length (the lower limit for NetMHCIIpan is 15 residues) and were cleaved at linkages between hydrophobic amino acids, which simulates the cathepsin degradation of antigenic proteins inside antigen processing cells [50,51]. The Bd-related peptide dataset comprised a total of 643 peptides (15–24 amino acid in length) from Bd 70 kDa heat shock protein 3, 26S protease regulatory subunit 6A-B, which are both virulence factors [52], and JEL423 surface antigen [53], as well as from gene classes (including proteases, adhesion, and lipase-3) with increased host expression after Bd exposure [54]. Additionally, we used two control datasets: (i) non-Bd fungal peptides—49 peptides derived from 15 ‘non-Bd-related’ immunogenic antigens from *Aspergillus fumigatus* [55], a major fungal pathogen in humans; and (ii) bacterial peptides—24 peptides from 7 *Escherichia coli* antigens. See Appendix A for details of the three datasets.

*Anuran MHC-II:* we used the same MHC-II alleles as those used for supertyping analysis above. Some alleles within the same species had the same binding affinity (BA; in nM) scores, irrespective of the peptide examined, and were therefore compiled into single unit variants for analyses. In total, after compilation, we collected BA scores for a total of 42 MHC-II variants from Japanese frogs, 15 from Korean frogs, and 96 from other frog species across the world.

NetMHCIIpan4.0 outputs the predicted measurements of BA for all 15mers within each of the peptides. Thus, in the cases where input antigen sequence was longer than 15 amino acids, the 15mer with lowest BA (i.e., stronger binding) was retained. Binding scores were then calculated as log-transformed binding affinity (log[BA]:$1-\log_{50k}\mathrm{BA}$); higher log[BA] is indicative of stronger binding. For analyses of prediction results, we surveyed the number of peptides that each MHC-II variant could bind with relatively strong affinity, based on a provisional threshold of log[BA] ≥ 0.3. In GraphPad Prism version 9.2.0 (GraphPad Software, San Diego, CA, USA), we used one-way analysis of variance to evaluate differences among supertypes, and Tukey’s multiple comparisons test to compare between supertypes.

### 2.6. Mitochondrial DNA (mtDNA) Sequences and Species Tree

We constructed a molecular phylogenetic tree using mtDNA sequences extracted from transcriptomic data to obtain an overview of species relationships. This was performed first by collating published sequences of 12 mitochondrial genes (ND1, ND3, ND4, ND4L, ND5, ND6, COX1, COX2, COX3, ATP6, ATP8, and CYTB) from 50 species across all major anuran families (non-anuran *Andrias davidianus* was included as outgroup). These 12 genes were used because they could be isolated following NCBI-BLAST search on assembled transcripts from the 11 Japanese study species (Appendix A). For five of our target species, both sequences isolated from transcript data as well as published mitochondrial genome sequences were included (*B. buergeri*, *G. rugosa*, *B. japonicus*, *P. nigromaculatus*, and *D. japonicus*). Sequences from the 12 genes were concatenated (total length 9709–10,329 bp) and aligned using BioEdit, and a neighbor-joining tree (p-distance, complete deletion) with 1000 bootstrap replicates was constructed using MEGA X [24].

## 3. Results

### 3.1. MHC Sequence and Phylogenetic Analyses

Using assembled transcriptomic data supplemented with molecular cloning and sequencing data, we characterized MHC variants from one individual from each of eight Japanese frog species, identifying 1–8 class I variants and 2–4 class II variants per species (Table 2, Figure 1). The minimum number of loci varied more for MHC class I (1–4 loci/individual) than class II (1–2 loci/individual), which is concordant with our previous characterization in multiple individuals in three Japanese *Rana* frogs (2–5 MHC-I loci [23] and 1–3 MHC-II loci [21] per individual). Sequences of all newly identified class I and IIb alleles have been submitted under GenBank accession numbers OQ473758–OQ473791 and OQ473792–OQ473811, respectively.

Our phylogenetic analyses of MHC-I α1–2 domain and MHC-II β1 domain sequences (Figure 2) revealed evidence of trans-species polymorphism, which is a common characteristic of MHC. For example, MHC-I variants Drja-UA*04, Peni-UA*05, Pepo-UA*02, Feja-UA*04, and Glru-UA*03 did not form distinct species-specific clusters with the remaining variants of the respective species and were even separated beyond higher taxonomic groupings (Figure 2A). For MHC-II, variants seemed to cluster within families, with the exception of a separate branch, including 71 variants from *L. yavapaiensis* (Rajidae), 101 variants from *Z. omeimontis* (Rhacoporidae), 10 variants from a Chinese population of *P. nigromaculatus*, and 2 variants from the Japanese *P. nigromaculatus* individual identified in the present study. This separate phylogenetic grouping is suggestive of at least two distinct MHC-II loci that are relatively divergent among anuran frogs. This distinction is also supported by amino acid sequence alignment (Figure 1B) and supertyping results below.

**Figure 2 animals-13-02121-f002:**
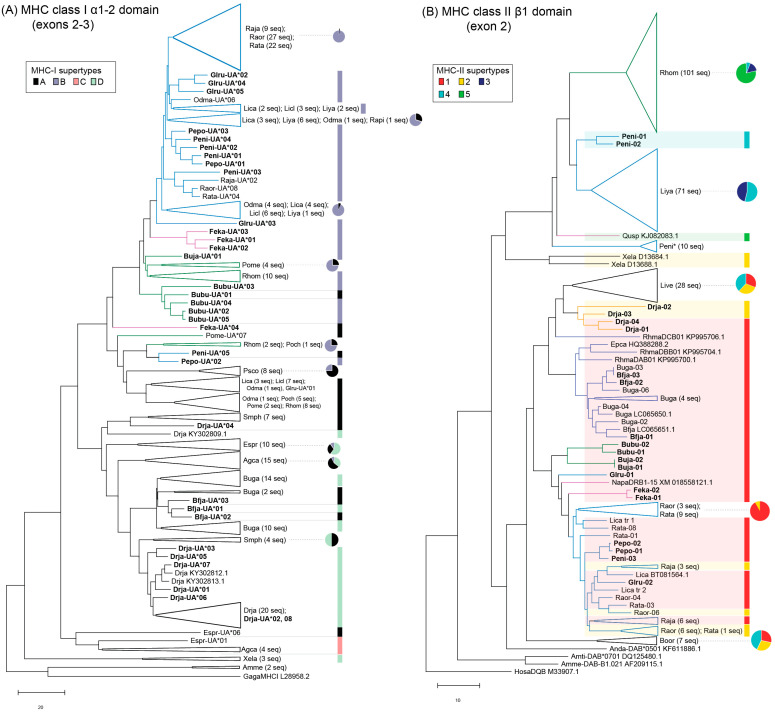
Phylogenetic relationships among (**A**) MHC class I α1 + α2 or (**B**) MHC class II β1 amino acid variants identified in Japanese anurans and other amphibian species using a neighbor joining method based on putative neutral sites (non-synonymous substitutions at the non-PBR sites + all synonymous substitutions). Chicken MHC-I (L28958.2) or human MHC-II (M33907) were used as outgroup sequences. MHC-I and/or MHC-II sequences from other species include: *Agalychnis callidryas* (Agca), *Ambystoma mexicanum* (Amme), *Ambystoma tigrinum* (Amti), *Andrias davidianus* (Anda), *Bombina orientalis* (Boor), *Bufo gargarizans* (Buga), *Epidalea calamita* (Epca), *Espadarana prosoblepon* (Espr), *Lithobates catesbeianus* (Lica), *Lithobates clamitans* (Licl), *Lithobates yavapaiensis* (Liya), *Litoria verreauxii alpina* (Livea) *Nanorana parkeri* (Napa), Odorrana margaretae (Odma), Polypedates megacephalus (Pome), *Pseudophryne corroboree* (Psco), *Quasipaa spinosa* (Qusp), *Rana pipiens* (Rapi), *Rhinella marina* (Rhma), *Zhangixalus (Rhacophorus) omeimontis* (Rhom), *Zhangixalus (Rhacophorus) chenfui* (Poch), *Smilisca phaeota* (Smph), *Xenopus tropicalis* (Xetr), and *Xenopus laevis* (Xela). Additional accession numbers are shown in Appendix A. Trees also include previously characterized MHC-I sequences from a Korean population of *Dryophytes japonicus* (Drja) and MHC-IIb sequences from a Chinese population *of Pelophylax nigromaculatus* (Peni*). Colored branches indicate the same families shown in the species tree in Figure 3. Supertypes allocated to MHC variants are indicated on the right of variant/branch labels; pie charts are used for variants within compressed groups/clusters. MHC variants from distantly related species were excluded from supertype analyses including MHC-II from Peni* because these alleles formed a distinct cluster in preliminary supertype analyses.

### 3.2. Species Tree

The species tree using the 12 mitochondrial gene sequences confirmed the appropriate phylogenetic relationships of the 11 Japanese study species among other frogs (Figure 3). Moreover, the five Japanese species with previously available sequences had high bootstrap similarity with sequences collated in the present study. Data from the 11 study species have been deposited under accession number PRJDB11134 in the NCBI BioProject database (https://www.ncbi.nlm.nih.gov/bioproject/). TimeTree [56] was used to estimate the evolutionary divergence times between the anuran families of interest, by inputting two of the study species in pairwise fashion. For example, the estimated divergence time for the ancestor of all anuran families found in Japan (i.e., Ranidae, Rhacophoridae, Dicroglossidae, Hylidae, and Bufonidae) is 155 MYA (confidence interval 134.1–160.0 MYA).

### 3.3. MHC Supertyping Analyses

MHC-I variants from the 11 Japanese frog species sequenced in this study, along with 14 other species, were assigned to one of four class I supertypes (ST-A to ST-D; Figure 4A and Appendix A). Unsurprisingly, the supertype groupings tended to reflect the relationships among variants based on both sequence similarities and phylogenetics (Figure 1A and Figure 2A). For example, phylogenetically similar MHC-I variants from *A. callidryas* and *E. prosoblepon* formed the minor supertype ST-C. Among the Japanese frogs, most variants from Ranidae, Rhacophoridae, and Dicroglossidae were assigned to ST-B, whereas *B. japonicus* (Bufonidae) and *D. japonicus* (Hylidae) variants were mainly assigned to ST-A and ST-D (Figure 4A, Appendix A). This separation of MHC-I supertypes is also in concordance with the large evolutionary divergence time between Ranidae/Rhacophoridae/Dicroglossidae and Bufonidae/Hylidae of 155 MYA (confidence interval 145–160 MYA) (Figure 3) according to TimeTree estimations. The distribution of MHC-I supertypes skewed depending on Bd-susceptibility (chi-square *p*-value < 0.001; Appendix A), although this was based on generalized or assumed susceptibility information. In particular, the 69 alleles in ST-D all belong to species that are assumed to be resistant or tolerant to Bd. This includes 3 alleles from *X. laevis*, several alleles from East Asian *D. japonicus* and *Bufo* spp., and 13 alleles from the three Central American species examined. Although MHC-I data from ‘Bd-susceptible’ anurans was limited (18 alleles from three species), alleles were assigned to ST-A or ST-B (Appendix A).

MHC-II variants from the Japanese frogs and other frogs from across the world were assigned to one of five class II supertypes (ST-1 to ST-5; Figure 4B, Appendix A). In a similar manner to MHC-I, the MHC-II supertype groupings followed with sequence alignment and phylogenetic results (Figure 1B and Figure 2B). The variants from a distinct phylogenetic grouping, which included two variants from Japanese *P. nigromaculatus*, were assigned to ST-3, ST-4, and ST-5. Interestingly, one common MHC-II supertype, ST-1, was shared among all 11 Japanese frog species examined in this study as well as two species found in Korea (*B. gargarizans* and *B. orientalis*), despite the large evolutionary divergence among some of the species; all of these 13 species are assumed to be Bd-resistant. The remaining alleles in ST-1 comprise allele(s) from *N. parkeri*, *E. calamita*, the Bd-resistant carriers *L. catesbeianus* and *R. marina*, and nine alleles from the susceptible species *L. verreauxii alpina*, of which four were found in Bd-resistant/surviving individuals (the remaining four alleles from ‘Bd-resistant’ *L. verreauxii alpina* were assigned to ST-4) [16].

**Figure 3 animals-13-02121-f003:**
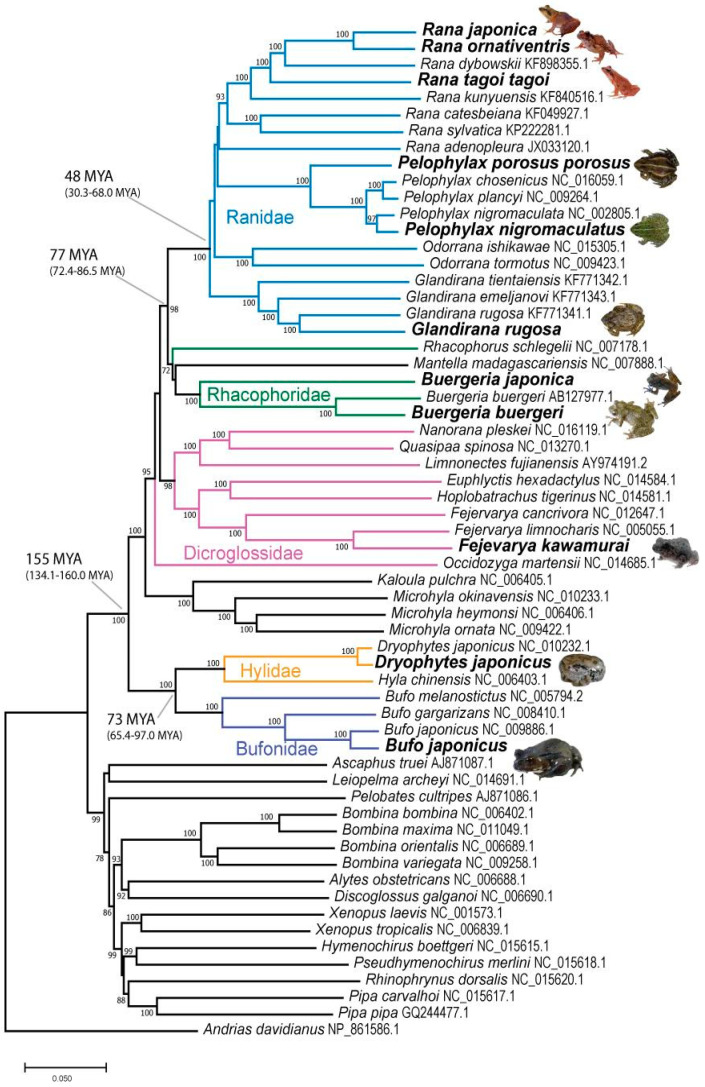
Species tree based on mitochondrial control gene sequences. Bold indicates sequences collated from Japanese frog species in the present study. For all other sequences, accession numbers are indicated next to species names. Colored branches and labels indicate the five different anuran families from which frogs were sequenced for this study. Divergence times (and confidence intervals) are based on estimates from the TimeTree database. Photo sources: Q. Lau.

**Figure 4 animals-13-02121-f004:**
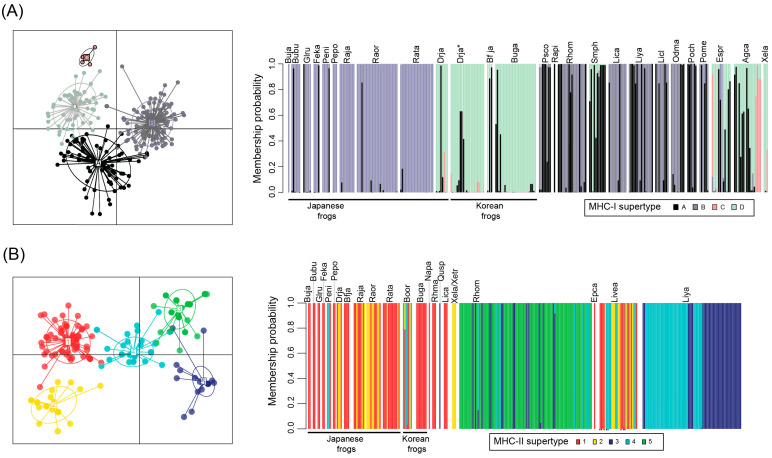
Supertype clustering and composite plots of (**A**) MHC-I and (**B**) MHC-II variants from Japanese and other anuran species. For class I, four supertypes were grouped (ST-A to ST-D); for class II, five supertypes were grouped (ST-1 to ST-5). See Figure 2 caption for species names. * indicates MHC-I from Korean population of *D. japonicus* (Drja). Small ‘R’ below MHC-II from *L. verreauxii alpina* (Livea) indicate ‘resistant’ alleles.

### 3.4. MHC-II Binding Prediction

The predicted binding affinity (i.e., log[BA] scores) estimated for 151 MHC class II variants from frogs across the world against the Bd-related peptides largely varied between variants and frog species (Appendix A). After grouping the predicted binding affinity by MHC-II supertype, we found that ST-1 and ST-2 bound a significantly higher number of Bd-related peptides with good affinity (i.e., log[BA] ≥ 0.3) than ST-3 and ST-5 (*p* < 0.0001) and ST-4 (*p* < 0.05) (Figure 5A; adjusted *p*-values). However, this finding was not exclusive to Bd-related peptides; the same pattern also applied to non-Bd fungal peptides and bacterial peptides (i.e., higher number of peptides with log[BA] ≥ 0.3 in ST-1 and ST-2) (Figure 5B,C).

## 4. Discussion

In the current study, we generated transcriptomic resources from a variety of Japanese frog species, most of which lack any genomic resources. Comparison of the MHC from Japanese frogs, assumed to be Bd-resistant, with other frogs worldwide with variable susceptibility to Bd revealed some notable findings. MHC supertyping analyses identified class I and class II supertypes of interest; in particular, MHC class II ST-1 comprised alleles from every Japanese and Korean anuran investigated, despite the large evolutionary divergence among species. In addition, MHC binding prediction analyses provided preliminary evidence that ST-1 along with ST-2 may have overall higher peptide binding ability to Bd-related peptides as well as non-Bd fungal peptides and bacterial peptides.

The transcriptomic dataset from eight Japanese anurans, along with our previous transcriptome report of three Japanese *Rana* species, provides a framework for investigating immune genes of Japanese anurans. We focused on utilizing the transcriptomic data of the various Japanese species to study the highly polymorphic MHC class I and II genes and extending the understanding of the mechanisms and genetic basis of resistance in East Asian anurans to chytridiomycosis.

Despite the growing number of studies associating MHC genotypes with Bd resistance, direct causal evidence is currently lacking. For example, in *L. yavapaiensis*, the role of MHC in chytridiomycosis resistance remains relatively unclear, with both MHC-II heterozygous allele state as well as a specific allele being linked to a reduced risk of death after experimental Bd infection [18,19]. In another study, conformation at the MHC-II binding pocket 9 was shared among Bd-resistant Korean frogs and was also more frequent in surviving Australian *L. verreauxii alpina* individuals that were exposed to Bd infection [16]. The present study expands on the knowledge of host-chytridiomycosis dynamics and continues to consider that frogs in East Asian and other anuran populations across the world may have unique features in MHC function, whether it be specific physicochemical properties of the peptide binding region or overall peptide binding capacity, that contribute to resistance against infectious pathogens such as Bd.

One of the major findings of our study is the identification of an MHC-II supertype (ST-1) that is shared among all Japanese and Korean frog species considered. The analyses of mitochondrial genes indicate that some of these frog species in Japan diverged from a common ancestor approximately 150 million years ago. Despite this, the conformation of MHC class II binding pockets in these frogs has either been conserved over evolutionary time or evolved via convergent evolution by the same selective force. This is not unexpected given that many of the major evolutionary forces that drive selection in adaptive genes, and even speciation, include environmental factors and infectious diseases.

Moreover, ST-1 is not confined to one geographical region; it comprised MHC-II alleles from non-Asian frogs including *L. catesbeianus* and *R. marina*, which are both known to be carriers or reservoirs of Bd [39,40]. The former is an invasive species in Japan and was reported to have the highest Bd haplotype diversity within the country [15]. The predicted peptide binding ability of most MHC-II alleles from these two species, along with most other alleles in ST-1, is generally high (see Appendix A). Even though there is no noticeable distinction in MHC-II function between Bd-resistant and Bd-tolerant/carrier species, the combination of MHC-II supertyping and binding predictions provides a platform for more targeted region- or species-specific studies in the future.

MHC-II ST-1 in our supertyping analysis also comprised nine alleles from Bd-susceptible *L. verreauxii alpina* from Australia, four of which were from ‘Bd-resistant’ individuals (i.e., survivors after experimental infection) [16]. Alleles from this species diversify among ST-1, ST-2, and ST-5, the latter of which contains the remaining ‘resistant’ alleles (Appendix A). This suggests that some *L. verreauxii alpina* MHC alleles may be shaped by similar selective forces to that in Bd-resistant frogs from Asia and the Americas. Nevertheless, since the identification of ‘resistant’ alleles was assigned based on the presence of the alleles in a few *L. verreauxii alpina* individuals that survived Bd infection [16], large-scale controlled infection-MHC genotyping studies will be important in the future to better understand the impacts of these putative Bd-resistance alleles.

We note that, apart from three Ranidae frogs extensively studied previously [21,23], the Japanese study species for MHC characterization, supertype analyses, etc. are based on single individuals of each species. This limits us from comprehensively assessing MHC intraspecies allelic variation and immune function. Nevertheless, our data from single individuals of study species still identified an MHC-II supertype shared among all Japanese species, and increased sample size in the three Ranidae frogs (*n* = 7 per species) did not reveal any additional class I or II supertype diversity (Figure 4). In any case, the present study has established primer sequence information that can be utilized for high throughout MHC sequencing of multiple individuals and comprehensive supertype cluster analyses in the future.

From MHC-I supertyping analyses, we found that one supertype (ST-D) only comprised alleles from species that were assumed to be Bd-resistant (including two Japanese anurans) or reported to be quite tolerant. However, there is relatively limited class I allele data available from Bd-susceptible anurans. Thus, further MHC-I genotyping of additional susceptible species or individuals is required to determine whether there are any functional properties related to MHC class I supertypes.

We attempted to predict the binding capacity of anuran MHC class II alleles against peptides derived from Bd. No obvious trends were identified apart from two supertypes with overall higher binding affinity function irrespective of antigen or peptide origin. ST-1, which is shared amongst all Korean and Japanese frogs studied, is one of these two supertypes. It remains possible that ST-1 (and ST-2) have certain physicochemical properties that allow them to recognize a high diversity of antigenic fragments, and subsequently have a more diverse immune repertoire. However, the lack of full-length MHC-II sequences (especially for some previously published anuran species) for the pan-specific prediction algorithm may have limited the accurate prediction of binding affinity. Nevertheless, this is one of the first studies to report the application of an MHC binding prediction algorithm in a non-human non-model species. Our preliminary findings using this algorithm may provide a springboard for further characterizations of the Bd-binding antigen capacities of anuran MHC-II receptors, including in vitro assays, to experimentally measure the binding affinity of recombinant host MHC-II proteins to processed or recombinant Bd peptides.

Although resistance to most diseases is polygenetic, the MHC is an ideal candidate gene region for preliminary elucidation of relationships between anuran hosts and pathogens, including amphibian chytrid fungus, due to the associations previously reported between MHC alleles and chytrid resistance in frogs and its importance for disease resistance in other species. In addition, our transcriptome datasets provide a resource to investigate other candidate genes that may contribute to disease resistance.

Even though East Asian frogs are mostly resistant to Bd, the current global distribution of the pathogen and the host-pathogen arms race may cause diversification and increased virulence of Bd to East Asian frogs. Indeed, new associations have been recently identified between host anuran species and distinct Bd lineages in North America [57]. Thus, in the future, the resistant frogs in East Asia may potentially face new emerging Bd lineages that are more virulent or can evade host immunity of these frogs. As with all host-pathogen interactions, the anuran-chytridiomycosis relationship is dynamic; changes to this relationship can bring forth opportunities for pathogen spread or evolution. For example, there may be the risk of new hybrid Bd lineages forming [58] as globalization and the wildlife pet trade continues without adequate biosecurity.

Tip-dating analysis predicted that the ancestor of BdGPL, the most pathogenic Bd lineage that is responsible for high mortality of frogs worldwide, originated 120–50 MYA [12]. This overlaps with the predicted divergence times of some of the Japanese species investigated in the present study. Due to the ancient origin of Bd, speciation within Japan and East Asia would have likely occurred amongst the presence of the ancestor of this fungus. While it is not known whether the ancient lineage of Bd strains were pathogenic, chytrid infections have been detected in Asian amphibians as early as the 20th century [15,59]. Our finding of an MHC-II supertype that shows broad apparent binding affinity to bacteria and fungi and is shared amongst all the examined East Asian anurans, suggests that MHC-II function (e.g., binding affinity to Bd and other pathogens) may have been conserved since Japanese frogs diverged up to ~150 MYA. If that is the case, then it may be possible that Bd has infected East Asian frogs ever since the ancestral lineage emerged, leading to frogs in the region evolving resistance or co-adapting with Bd.

## 5. Conclusions

Using transcriptomics data with additional sequencing, we characterized MHC class I and II genes from a range of Japanese frogs spanning an evolutionary divergence of up to ~150 million years. East Asian frogs are seemingly resistant to the deadly Bd; this is likely because this fungus originated from the Asian region. From comparisons with available MHC allele sequences from other anurans that have variable Bd susceptibility, we identified MHC supertypes that may be related to Bd resistance. In particular, MHC-II ST-1 included alleles from all Japanese and Korean frogs examined in this study. The role of this supertype in host-chytridiomycosis dynamics requires further elucidation, but preliminary MHC predicted binding suggests that this supertype is one of two supertypes with generally higher fungal and bacterial binding capacity.

## Figures and Tables

**Figure 5 animals-13-02121-f005:**
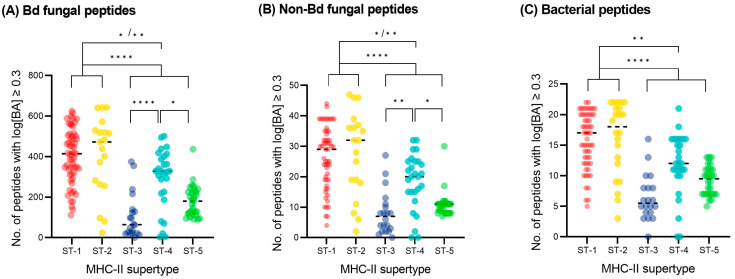
Predicted binding of anuran MHC-II to (**A**) Bd-related peptides, (**B**) non-Bd fungal peptides (from *Aspergillus fumigatus*), and (**C**) non-fungal bacterial peptides (from *Escherichia coli*), grouped by MHC-II supertype. Each dot represents the number of peptides bound with high binding affinity (i.e., above the provisional threshold of log[BA] ≥ 0.3) by a single MHC-II variant. Comparisons between supertypes were performed using Tukey’s multiple comparisons test: **** *p* < 0.0001; ** *p* < 0.01; * *p* < 0.05.

**Table 1 animals-13-02121-t001:** Summary of primers used in this study.

Target Gene	Target Species	Forward Primer (5′–3′)	Reverse Primer (5′–3′)	Amplicon Length (bp)
MHC class I	All study species	GAYRGHCACWBYYTSCGBTAYT	CTCYGGRCGYACTCTCYTYT	552–555
MHC class II	*Buergeria* spp., *Glandirana rugosa*, *Pelophylax* spp.	RVATTAYHWGCAGMAASATG ^a^	CACWCCRGCAAYRATAARYA ^a^	752–761
	*Fejervarya kawamurai*	AGGAGAAKCCGCTGATTATG	CACAGSTGAAGDYRTCTCCTTTYK	524
	*Pelophylax nigromaculatus*	TACTATCCGCCTCCCATCC	GAGCCCAACAATGAAGAAA	663
	*Dryophytes japonicus*	GYTCSTCACCAGATGTCAGA	CCAGGATCTGGWAAGTCCAR	486–489
	*Bufo japonicus* ^b^	(i) CTTTCTGAACGGGACTCAGC (ii) AAGCCGAGAACCAGAAGACA	(i) YTSRACTCTCCGGTCWGYWR (ii) MGAKGGGTAGAARCCAARCA	244 466–469

^a^ Primers established in our previous study [21]. ^b^ The 1st shorter primer set was used to amplify and confirm one of the *B. japonicus* MHC IIb variants that was present as an assembled contig but could not be amplified by the 2nd longer primer set.

**Table 2 animals-13-02121-t002:** Number of MHC class I and II variants identified from transcriptome and molecular cloning of one individual per species. Values in parentheses indicate minimum number of loci. Refer to [21,23] for three previously studied Japanese *Rana* species.

Species	No. of MHC-I Variants	No. of MHC-II Variants
*Buergeria buergeri* [Bubu]	4 (2)	2 (1)
*Buergeria japonica* [Buja]	1 (1)	2 (1)
*Bufo japonicus* [Bfja]	3 (2)	3 (2)
*Dryophytes japonicus* [Drja]	8 (4)	4 (2)
*Fejervarya kawamurai* [Feka]	4 (2)	2 (1)
*Glandirana rugosa* [Glru]	5 (3)	2 (1)
*Pelophylax nigromaculatus* [Peni]	4 (2)	3 (2)
*Pelophylax porosus porosus* [Pepo]	3 (2)	2 (1)

## Data Availability

Transcriptome data are deposited in DDBJ under accession number DRA016614. Accession numbers for all MHC and microsatellite sequences are provided in the manuscript.

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
