# Peer review of "Conserved Evolution of MHC Supertypes among Japanese Frogs Suggests Selection for Bd Resistance"

_animals, 2023, doi:10.3390/ani13132121_

Round 1
Reviewer 1 Report
Lau et al. present a thorough study examining the evolutionary origins of Bd-resistance in frogs from East Asia. Novel transcriptomes were assembled from eight frog species for the purpose of creating/grouping MHC supertypes, from which phylogenetic and predictive functional analyses were performed. The most interesting finding concerned the MHC-II supertypes: two of the five supertypes contained alleles that were common among the Bd-resistant species, despite relatively distant phylogenetic relationships. These supertypes were also predicted to express significantly higher peptide-binding ability, which could represent a conserved, broad-spectrum immunological mechanism of resistance to pathogens. Consequently, this study offers insight into the complex evolutionary history between B. dendrobatidis and its host taxa.
This study is informative for the reasons described above, but it is quite dense. The manuscript and its approach build upon previous work conducted by the authors; consequently, the methods are touched upon only briefly, which makes it difficult to follow for someone not familiar with the seminal studies. That being said, the authors were diligent in referencing the sources of this information and in explaining the general rationale of their approach. Outside of a few minor suggestions, I believe that the submitted manuscript would be a valuable addition to this journal.
Minor Concerns:
The scope of this study is broad, and the methods are high throughput; however, the approach makes some equally broad assumptions, as all large-scale studies must do. I do not question the value of the study, but I would like for the authors to better explain their rationale for the assumptions listed below:
§ The conclusions of this study are based on transcriptomic comparisons between a single representative of each species. I am aware of the logistical constraints associated with -omic approaches of this magnitude, which often necessitate smaller sample sizes. Nevertheless, such an approach does not account for intraspecific allelic diversity in MHC or other immunological properties, which could conceivably alter the outcome of the supertype clustering analysis.
§ This study provided further support for the role of MHC in Bd resistance that were evidenced by geographical patterns in expression. This conclusion, however, is based on subjective classifications of Bd susceptibility (e.g., susceptible, tolerant, resistant) that were compiled from literature. It is unclear whether these classifications were based on common criteria within the field, or if the authors categorized them based on their own judgement before performing the analysis.
Minor Suggestions:
If it does not conflict with journal formatting preferences and the authors agree, I suggest that Figure S2 be integrated into Figure 5. I understand that the focus of the protein kinetics modeling is on Bd-related peptides, but it is striking to see the same patterns of binding affinity among MHC supertypes in a side-by-side comparison. Each of the three plots are small and could be presented as individual panels within Figure 5. Depicting all three in the main text supports the conclusion that molecular mechanisms of Bd-resistance may be an example of evolutionary co-option from a much older and more ubiquitous immunological threat.
Line-by-line feedback:
Supplemental Information: Fix spelling error in plot title of Figure S2B.
Author Response
Dear Reviewer 1,
Thank you for your time and effort to review our manuscript. Below is our point-by-point response to your comments. Responses are in blue color.
- The conclusions of this study are based on transcriptomic comparisons between a single representative of each species. I am aware of the logistical constraints associated with -omic approaches of this magnitude, which often necessitate smaller sample sizes. Nevertheless, such an approach does not account for intraspecific allelic diversity in MHC or other immunological properties, which could conceivably alter the outcome of the supertype clustering analysis.
Thank you for pointing this out. We have now added a small paragraph to note that the MHC sequences are mainly from one individual per species and this may impact the results and analyses.
NEW TEXT: “We note that, apart from three Ranidae frogs extensively studied previously [21,23], the Japanese study species for MHC characterization, supertype analyses, etc. are based on single individuals of each species. This limits us from comprehensively assessing MHC intraspecies allelic variation and immune function. Nevertheless, our data from single individuals of study species still identified a MHC-II supertype shared among all Japanese species, and increased sample size in the three Ranidae frogs (n = 7 per species) did not reveal any additional class I or II supertype diversity (Figure 4). In any case, the present study has established primer sequence information that can be utilized for high throughout MHC sequencing of multiple individuals and comprehensive supertype cluster analyses in the future.”
- This study provided further support for the role of MHC in Bd resistance that were evidenced by geographical patterns in expression. This conclusion, however, is based on subjective classifications of Bd susceptibility (e.g., susceptible, tolerant, resistant) that were compiled from literature. It is unclear whether these classifications were based on common criteria within the field, or if the authors categorized them based on their own judgement before performing the analysis.
Thank you for your comment. We have now revised this sentence to emphasise that the categorization is based on our judgement of the literature before performing analyses.
REVISED TEXT “Additional Bd-susceptibility categorization for the species shown below is generalized based on extrapolation from previous reports [36–40], and used in the present study to examine general associations between MHC supertypes and Bd susceptibility.”
We do acknowledge that this is not the ideal classification approach, but nonetheless our discussion in the original manuscript has mentioned there are limited MHC data from Bd-susceptible species (especially for class I) that warrants genotyping of additional species in the future.
- If it does not conflict with journal formatting preferences and the authors agree, I suggest that Figure S2 be integrated into Figure 5. I understand that the focus of the protein kinetics modeling is on Bd-related peptides, but it is striking to see the same patterns of binding affinity among MHC supertypes in a side-by-side comparison. Each of the three plots are small and could be presented as individual panels within Figure 5. Depicting all three in the main text supports the conclusion that molecular mechanisms of Bd-resistance may be an example of evolutionary co-option from a much older and more ubiquitous immunological threat.
Thanks for your comment. Following suggestions from both you and Reviewer 2, we have integrated the figures from Figure S2 into Figure 5.
- Supplemental Information: Fix spelling error in plot title of Figure S2B.
Thanks for pointing out our spelling error. We have now amended the plot title for the figure (now Figure 5C)
Reviewer 2 Report
The manuscript entitled "Conserved evolution of MHC supertypes among Japanese frogs suggests selection for Bd resistance" by Lau et al. aims to expand the study of MHC genes across a diverse range of common (widely distributed) Japanese frog species and compare them to MHC from Batrachochytrium dendrobatidis-resistant and susceptible species, which will further contribute to understanding amphibian-chytridiomycosis dynamics. In general, the manuscript is really good. I would only suggest some minor revisions:
Introduction
- I suggest to add some sentences on what type of cells express MHC (focalizing on APCs)
- I would also add a paragraph on the evolutionary conservation of MHC positive cells among the different classes of vertebrates.
Some minor grammar and spell check are required.
Author Response
Dear Reviewer 2,
Thank you for taking the time and effort to review our manuscript and provide comments. Below is our point-by-point response to your comments. Responses are in blue color.
- I suggest to add some sentences on what type of cells express MHC (focalizing on APCs)
Thanks for your suggestion. We have now added information to indicated what types of cells express MHC .
REVISED TEXT: “MHC-I molecules are ubiquitously expressed on most nucleated cells and predominantly present endogenous antigenic peptides from intracellular pathogens (e.g. viruses); the α1 and α2 domains (heavy chain) form the peptide binding platform and surface for recognition by cytotoxic CD8+ cells [1,2]. MHC-II molecules are found on professional antigen-presenting cells (including B lymphocytes, dendritic cells, macrophages, and in most species, activated T lymphocytes) and present exogenous antigenic peptides from extracellular pathogens (e.g. many bacteria and fungi) to CD4+ T helper cells.”
- I would also add a paragraph on the evolutionary conservation of MHC positive cells among the different classes of vertebrates.
Thank you for your suggestion. We're sorry, we do not entirely understand your comment. Perhaps you are referring to the evolutionary conservation of MHC-I rather than cellular expression, since MHC-I is nearly ubiquitously expressed (which we have now indicated in the revised manuscript). Nevertheless, we feel that the concept of MHC expression and its potential conservation does not fit directly with the present manuscript. Thus, whilst we appreciate your suggestion, we would like to refrain from this addition.
Reviewer 3 Report
Genetic variations in the major histocompatibility complex (MHC) have been associated with resistance to the chytrid fungus Batrachochytrium dendrobatidis (Bd) in frogs in East Asia and worldwide.
The authors of the submitted manuscript isolated and validated class I and IIb MHC sequences from transcriptomic data of 11 Japanese frog species. They then compared the MHC of Japanese frogs with other species worldwide that are differentially sensitive to Bd. An analysis grouping the MHC alleles based on the physicochemical properties of the peptide binding sites revealed that all the East Asian frogs studied had at least one MHC IIb allele belonging to the ST-1 supertype. This suggests that despite the long divergence time between Japanese frogs, certain functional properties of MHC-II peptide binding sites have been conserved. Preliminary analyses using NetMHCIIpan-4.0 suggest that MHC-IIb ST-1 (and ST-2) have a higher overall peptide binding capacity than other supertypes, regardless of the origin of the peptides.
The authors conclude that MHC-IIb may have evolved in East Asian frogs under the same selection pressure and that Bd may be an important driver of MHC evolution in these amphibians.
The work seems solidly done and has produced interesting data. From my point of view, the manuscript is worthy of publication after two minor changes.
· It would be very desirable for the reader of the paper if both subfigures in Figure 4 (MHC I & II) are in color (this holds also true for Figures 1 and 2). The different shades of gray used are difficult to distinguish.
· Figures 5 and S2 should be combined into one figure for clarity. Although the text of the manuscript mentions (...MHC-IIb ST-1 and ST-2 have an overall higher peptide binding ability than other supertypes), one might get a flat impression of the respective results because the complete analyses (including S2) put the “specific abilities” of the two MHC-IIb supertypes (ST-1 & 2) into the right perspective.
Author Response
Dear Reviewer 3,
Thank you for reviewing our manuscript and positive feedback. Below is our point-by-point response to your comments. Responses are in blue color.
- It would be very desirable for the reader of the paper if both subfigures in Figure 4 (MHC I & II) are in color (this holds also true for Figures 1 and 2). The different shades of gray used are difficult to distinguish.
Thank you for your comment; we have now added color to the MHC class I supertyping results. These colors are reflected across Figures 1, 2, and 4.
- Figures 5 and S2 should be combined into one figure for clarity. Although the text of the manuscript mentions (...MHC-IIb ST-1 and ST-2 have an overall higher peptide binding ability than other supertypes), one might get a flat impression of the respective results because the complete analyses (including S2) put the “specific abilities” of the two MHC-IIb supertypes (ST-1 & 2) into the right perspective.
Thank you for your suggestion. Reviewer 2 also made a similar suggestion, so we have transferred Figure S2 to become a part of Figure 5.